# Regulation of RORα Stability through PRMT5-Dependent Symmetric Dimethylation

**DOI:** 10.3390/cancers16101914

**Published:** 2024-05-17

**Authors:** Gaofeng Xiong, Brynne Obringer, Austen Jones, Elise Horton, Ren Xu

**Affiliations:** 1Markey Cancer Center, University of Kentucky, Lexington, KY 40536, USA; ren.xu2010@uky.edu; 2Department of Pharmacology and Nutritional Sciences, University of Kentucky, Lexington, KY 40536, USA; 3Department of Veterinary Biosciences, The Ohio State University, Columbus, OH 43210, USA; 4Comprehensive Cancer Center, The Ohio State University, Columbus, OH 43210, USA; 5College of Arts and Sciences, The Ohio State University, Columbus, OH 43210, USA; obringer.46@buckeyemail.osu.edu (B.O.); jones.7544@buckeyemail.osu.edu (A.J.); 6Department of Food, Agricultural and Biological Engineering, The Ohio State University, Columbus, OH 43210, USA; horton.407@buckeyemail.osu.edu

**Keywords:** mammary epithelial cells, PRMT5, RORα, methylation, stability

## Abstract

**Simple Summary:**

Retinoid-related orphan receptor alpha (RORα), a member of the orphan nuclear factor family, is considered a potential tumor suppressor. Our previous studies have shown that a loss of RORα is associated with enhanced breast cancer malignancy, cell invasion and proliferation, epithelial–mesenchymal transition, and cancer-associated inflammation. The mechanisms of how RORα expression is regulated in mammary epithelial cells remain incompletely understood. In this study, we revealed a direct interaction between RORα and protein arginine N-methyltransferase 5 (PRMT5), which symmetrically dimethylated the DNA-binding domain of RORα and stabilized the RORα protein. The RORα protein was decreased in PRMT5-silenced mammary epithelial cells, accompanied by enhanced invasion and migration abilities. These findings uncover a novel mechanism for RORα regulation through PRMT5-induced symmetric dimethylation in breast epithelial cells.

**Abstract:**

Retinoic acid receptor-related orphan receptor alpha (RORα), a candidate tumor suppressor, is prevalently downregulated or lost in malignant breast cancer cells. However, the mechanisms of how RORα expression is regulated in breast epithelial cells remain incompletely understood. Protein arginine N-methyltransferase 5 (PRMT5), a type II methyltransferase catalyzing the symmetric methylation of the amino acid arginine in target proteins, was reported to regulate protein stability. To study whether and how PRMT5 regulates RORα, we examined the direct interaction between RORα and PRMT5 by immunoprecipitation and GST pull-down assays. The results showed that PRMT5 directly bound to RORα, and PRMT5 mainly symmetrically dimethylated the DNA-binding domain (DBD) but not the ligand-binding domain (LBD) of RORα. To investigate whether RORα protein stability is regulated by PRMT5, we transfected HEK293FT cells with RORα and PRMT5-expressing or PRMT5-silencing (shPRMT5) vectors and then examined RORα protein stability by a cycloheximide chase assay. The results showed that PRMT5 increased RORα protein stability, while silencing PRMT5 accelerated RORα protein degradation. In PRMT5-silenced mammary epithelial cells, RORα protein expression was decreased, accompanied by an enhanced epithelial–mesenchymal transition morphology and cell invasion and migration abilities. In PRMT5-overexpressed mammary epithelial cells, RORα protein was accumulated, and cell invasion was suppressed. These findings revealed a novel mechanism by which PRMT5 regulates RORα protein stability.

## 1. Introduction

Within the orphan nuclear factor family, retinoic-acid-receptor-related orphan receptor alpha (RORα) [1,2] is pivotal in various physiological processes, including circadian rhythm, metabolism, transformation, cell differentiation and inflammation [3,4,5], and disease content like neurodegenerative illnesses and carcinoma [6,7,8,9]. RORα has been reported as a potential tumor suppressor in solid tumors, including breast cancer, lung cancer, melanoma, colon cancer, hepatocellular carcinoma, prostate cancer, and glioma [6,10,11,12,13,14,15]. In our previous studies, we have determined that poor clinical outcomes in breast cancer patients are linked to the downregulation of RORα. A decreased expression of RORα in breast cancer fosters invasion and proliferation, enhances epithelial–mesenchymal transition (EMT), and triggers cancer-associated inflammation in the tumor microenvironment [9,10,16,17]. Therefore, the prevention of RORα loss or downregulation is a pathway to prevent breast cancer initiation and progression. However, the mechanisms of RORα downregulation in breast cancer cells are not fully understood.

Protein expression can be regulated at the level of transcription, translation, and post-translation. Post-translational modification is a critical mechanism for the regulation of protein subcellular localization, activity, and stability. Post-translational modification is mediated by diverse enzymatic processes, including phosphorylation, acetylation, ubiquitination, SUMOylation, hydroxylation, glycosylation, and methylation. Belonging to the methyltransferase family, protein arginine N-methyltransferase 5 (PRMT5) functions as a type II methyltransferase. It catalyzes the symmetric methylation of the arginine amino acid within target proteins, spanning histones and non-histone proteins [18,19,20]. PRMT5 is involved in the regulation of various cellular processes, such as transcription, genome regulation, stem cell, differentiation, cell cycle, etc., under normal and disease content [18,19,20]. It has been reported that PRMT5 regulates many transcription factors and affects their stability by either increasing or decreasing their protein half-life, including hypoxia-inducible factor-1 alpha (HIF1α), MYC, sterol regulatory element-binding protein 1 (SREBP1), E2F Transcription Factor 1 (E2F1), and Krüppel-like factor 4 (KLF4) [21,22,23,24,25]. 

In this study, our data showed that RORα protein directly interacted with PRMT5 and was symmetrically dimethylated and stabilized by PRMT5. Our previous report discovered that RORα suppressed EMT by repressing the transcription of Snail, a key transcription factor in EMT. Here, we confirmed that RORα was regulated by PRMT5, which is functional in cell invasion, stemness, and EMT morphology, including changes in Snail expression. We identified a novel mechanism by which RORα protein level is regulated at post-translational level, which is mediated by PRMT5 symmetric dimethylation.

## 2. Materials and Methods

### 2.1. Two-Dimensional and Three-Dimensional Cell Culture 

For the tissue culture plastic 2D cell culture, human normal mammary epithelial cell MCF10A cells (from American Type Culture Collection) were cultured in Dulbecco’s Modified Eagle’s Medium/F12 (DMEM/F12) (Sigma Aldrich, St. Louis, MO, USA) supplemented with 5% horse serum (Gibco, Brooklyn, USA), 0.5 mg/mL hydrocortisone (Sigma Aldrich), 20 ng/mL epidermal growth factor (EGF) (PEPROTECH, Rosemont, USA), 10μg/mL insulin (Sigma Aldrich), 100 ng/mL cholera toxin (Sigma Aldrich), and 1% penicillin–streptomycin (Sigma Aldrich). HEK293FT cells (a generous gift from Mina J. Bissell, Lawrence Berkeley National Laboratory) were maintained in Dulbecco’s Modified Eagle’s Medium (DMEM) (Sigma Aldrich) supplemented with 10% fetal bovine serum (FBS) (Sigma Aldrich), 6 mM L-glutamine (Sigma Aldrich), 0.1 mM non-essential amino acids (Sigma Aldrich), 1 mM sodium pyruvate (Sigma Aldrich), and 1% penicillin–streptomycin (Invitrogen, Waltham, MA, USA). Both cell lines were maintained in a humidified incubator at 37 °C with 5% CO_2_. Plasmocin^TM^ (Invivo Gen, San Diego, CA, USA) was used for the elimination and prevention of mycoplasma contamination.

For the 3D laminin-rich extracellular matrix culture, cells were trypsinized from tissue culture plastic, seeded as single cells on a thin layer of Matrigel (Corning, Corning, NY, USA 100 μL per well of a 24-well culture plate), and supplemented with medium (DMEM/F12, 2% horse serum, 0.5 mg/mL hydrocortisone, 5 ng/mL EGF, 10 μg/mL insulin, 100 ng/mL cholera toxin, and 1% penicillin–streptomycin) containing 5% Matrigel. A total of 4.0 × 10^4^ MCF10A cells were seeded in each well of a 24-well culture plate. The medium was change every 2 days. 

### 2.2. Virus Preparation and Stable Cell Lines

Human RORα complementary DNA (cDNA) (two isoforms, RORα1 and RORα4) and the ligand-binding domain (LBD) and DNA-binding domain (DBD) of RORα were cloned into pCDH1 plasmid and used to generate the expression vectors pCDH1-RORα1-HA, pCDH1-RORα1-Flag, pCDH1-RORα4-Flag, pCDH1-LBD-Flag, and pCDH1-DBD-Flag. PRMT5 cDNA (Mammalian Gene Collection MGC Human PRMT5 Sequence-Verified cDNA from horizon, catalog ID: MHS6278-202829982) was cloned into the pCDH1 plasmid and generated the expression vectors pCDH1-PRMT5-HA and pCDH1-PRMT5-Flag. Two PRMT5 knockdown plasmids, shPRMT5-1 (clone ID: TRCN0000303446, target sequence GGCTCAAGCCACCAATCTATG) and shPRMT5-2 (clone ID: TRCN0000303447, target sequence CCCATCCTCTTCCCTATTAAG), were purchased from Sigma. HEK293FT cells were transfected with either gene expression vectors or shRNA vectors along with a packaging lentivector. FuGENE was used as a transfection reagent (Promega). After 48 h post-transfection, culture supernatants containing viral particles were harvested and used to infect human mammary epithelial cells. Cells that stably expressed genes or silenced genes were selected by at least 7 days of puromycin treatment.

### 2.3. Coimmunoprecipitation 

The PRMT5 expression plasmid pCDH1-PRMT5-Flag and pCDH1-RORα-HA plasmid-transfected HEK293FT cells or pCDH1-PRMT5-HA and pCDH1-RORα-Flag plasmid or transfected HEK293FT cells were lysed using an ice-cold hypotonic gentle lysis buffer (HGLB). The HGLB was prepared using 10 mM Tris-hydrochloric acid (HCl), pH 7.5, 2 mM ethylenediaminetetraacetic acid (EDTA), 10 mM sodium chloride (NaCl), protease inhibitor cocktails (EMD Millipore, Temecula, USA), and 0.5% Triton X-100. This was followed by incubation on ice for 10 min, and an accurate volume of high-concentration NaCl solution (5 M) was added to the cell lysis to achieve a final concentration of 150 mM NaCl. Anti-Flag M2 affinity gel (Sigma Aldrich) was used to pull down the protein complexes. The immunoprecipitated proteins were eluted by an elution buffer and denatured at 95 °C 10 min for Western blotting analysis.

### 2.4. GST Pull-Down Assay

The full-length PRMT5 gene was cloned into a pGEX-4T-1 bacterial vector for expressing the glutathione S-transferase (GST)-PRMT5 fusion protein. Expression of the GST-fused proteins was induced in Escherichia coli by adding 0.1 mM isopropyl-β-d-thiogalactopyranoside (IPTG) and incubating at 19 °C overnight with shaking. The bacteria were harvested and lysed using radioimmunoprecipitation assay (RIPA) buffer (50 mM Tris, 0.5% Na. deoxycholate,150 mM NaCl, 1% protease inhibitors, 1% NP-40), followed by purification using Pierce Glutathione Agarose from Thermo Scientific (Waltham, MA, USA). The expression of purified GST-fused proteins, GST-PRMT5, was confirmed by Coomassie staining after SDS gel electrophoresis. For the in vitro expression of RORα, HEK293FT cells were transfected with pCDH1-RORα1-Flag plasmid or pCDH1 vector control with FuGENE. The HEK293FT cells were lysed in RIPA buffer containing 1% protease inhibitors 36 h post-transfection. For the pull-down assay, GST control and GST-fused protein, GST-PRMT5, binding beads were incubated with RORα-expressing HEK293FT cell lysis at 4 °C for 4 h, followed by washing with RIPA buffer for three times. The pulled-down proteins were denatured at 95 °C 10 min for Western blotting analysis. 

### 2.5. Cycloheximide Chase Assay

RORα expression vector pCDH1-RORα-Flag plasmid and vector control/pCDH1-PRMT5-HA/shPRMT5 were transfected into HEK293FT cells cultured in a 12-well plate using FuGENE transfection reagent. About 48 h after transfection, cycloheximide (CHX, 100 μM) was added to the HEK293FT cells. Cell lysis samples were collected to examine the RORα protein levels at different time points (0, 1, 4, 8 h) after cycloheximide treatment using Western blotting.

### 2.6. Western Blotting

Cells cultured on plastic were lysed in laminine buffer (2% SDS in PBS buffer, 1% protease inhibitor cocktails from EMD Millipore). The protein concentration of the cell lysis was determined using a Pierce™ BCA Protein Assay Kit from Thermo Fisher (Waltham, MA, USA). Equal amounts of protein lysates were then loaded onto SDS gel for electrophoresis, followed by immunoblotting with primary antibodies at 4 °C overnight. Horseradish peroxidase- or DyLight 680/800- or StarBright Blue 520/700-conjugated secondary antibodies were used for imaging. Quantification of the Western blotting results was performed using Image J.JS analysis.

### 2.7. Real-Time PCR

Total RNA was isolated from cells using TRIzol reagent from Invitrogen, followed by cDNA synthesis with a SuperScript™ III First-Strand Synthesis kit (Invitrogen, Waltham, MA, USA) using a sample of 1.0 μg of the cells’ total RNA. The cDNA synthesis procedure was carried out according to the manufacturer’s protocol. A quantitative reverse transcription–PCRs (RT-PCRs) reaction was performed using SYBR green PCR master mix reagents (Applied Biosystems, Foster City, CA, USA) on a QuantStudio™ 3 Real-Time PCR System (Thermo Fisher). The thermal cycling conditions included an initial denaturation step at 95 °C for 30 s, followed by 40 cycles of amplification at 95 °C for 5 s, 55 °C for 30 s, and 72 °C for 15 s. The relative quantification of gene expression was determined using the △ threshold cycle (△CT) method. The following primer sequences were utilized for amplification: PRMT5, 5′-AGCCACTGCAATCCTCTTACTAT-3′ and 5′-TCAGGAAGATAACACCAACCTGG-3′, and 18S rRNA, 5′-ACCTGGTTGATCCTGCCAGT-3′ and 5′-CTGACCGGGTTGGTTTTGAT-3′. 

### 2.8. Invasion Assay

Transwell inserts (Corning) were precoated with 60 µL of Matrigel (1 mg/mL) and incubated at 37 °C for 30 min. Either 1 × 10^5^ cells of a vector control or shPRMT5 MCF10A cells suspended in 200 μL of medium were seeded onto the top of the Transwell filters, followed by incubation at 37 °C with 5% CO_2_ for 24 h. The cells that invaded through the Transwell filter were fixed by methanol at room temperature for 20 min. After staining with 8% crystal violet, images of the invaded cells on the bottom of the Transwell filters were captured using a Nikon microscope (Melville, NY, USA), and the number of invaded cells was quantified.

### 2.9. Single-Cell Migration Assay

Either a ector control or PRMT5-silenced MCF10A cells (4 × 10^4^) were seeded on collagen-I-pre-coated 35 mm glass-bottom dishes in DMEM/F12 medium supplemented with 2% FBS and EGF (4 ng/mL). After cell adhesion on the collagen-I-pre-coated glass-bottom dishes, and after about 2 h of incubation at 37℃, dynamic imaging of the cell movement was captured using a live cell/incubator imaging system (Nikon Biostation IMQ, Melville, NY, USA), with images taken at 10 min intervals over an 8 h period. 

### 2.10. Mammosphere Assay

Ultra-low-adherent plates were prepared by incubating with poly 2-hydroxyethyl methacrylate (poly-HEMA, 12 mg/mL in 95% ethanol) at 53 °C overnight. Cells cultured on 2D plastic dishes were detached using the standard trypsinization procedure and then filtered through a cell strainer (40 µm, Corning). The number of viable cells was determined using trypan blue staining and a hemocytometer. For the stem cell culture, each well of the ultra-low-adherent 24-well culture plates was filled with DMEM/F12 medium supplemented with EGF (20 ng/mL), B27 (1:50), insulin (5 μg/mL), basic fibroblast growth factor (20 ng/mL), gentamycin (100 µg/mL), and hydrocortisone (0.5 μg/mL). A single-cell suspension was then seeded at 1.0 × 10^4^ per well of the 24-well culture plates in triplicate. After 5 days of incubation at 37 °C with 5% CO_2_ without disturbance, mammospheres larger than 50 μm in diameter were counted under a microscope. The mammosphere-forming efficiency (MFE, number of mammospheres per well divided by the number of cells seeded per well multiplied by 100) ratio was calculated and quantified. 

### 2.11. Statistical Analysis

The experiments were conducted independently a minimum of three times. The results were presented as mean ± S.E.M., and the significance of the differences was evaluated using the independent Student’s *t*-test. A *p*-value of less than 0.05 indicated statistical significance, while a *p*-value of less than 0.01 represented a higher level of statistical significance.

## 3. Results

### 3.1. RORα Directly Interacts with PRMT5

The methylation of arginine residues is the predominant methylation manner in mammalian cells, which modifies the protein’s interacting properties [18]. PRMT5 is an enzyme reported to regulate protein stability by catalyzing the symmetric methylation of target proteins [21,22,23,24,25]. Our previous study showed that nuclear transcriptional factor RORα expression is high in the cell nucleus of normal mammary epithelial cells, in which PRMT5 mostly is localized in the nucleus [26,27]; RORα expression is downregulated in aggressive breast cancer cells, in which PRMT5 is mainly localized in the cytoplasm [26,27]. We hypothesize that PRMT5 interacts with RORα and regulates its stability in the cell nucleus. To test our hypothesis, we performed coimmunoprecipitation experiments to determine whether PRMT5 directly interacts with RORα; PRMT5-Flag and/or RORα-HA expressing plasmids were co-transfected in HEK293FT cells, and the protein complexes were immunoprecipitated by the anti-Flag antibody. A significant amount of RORα was pulled down in the RORα-HA-expressing cells but not in the control cells, indicating that PRMT5 bound to RORα in the HEK293FT cells (Figure 1a). To solidify our conclusion of the direct interaction between PRMT5 and RORα, we performed an in vitro GST pull-down experiment, and the result also showed that the RORα protein directly bound with the GST-PRMT5 protein purified from *Escherichia coli* (Figure 1b). The results of the coimmunoprecipitation and in vitro GST pull-down assay indicate that PRMT5 directly interacts with RORα to regulate its activity.

### 3.2. RORα Is Symmetrically Dimethylated and Stablized by PRMT5 

PRMT5 is an enzyme that symmetrically dimethylates in histone and non-histone substrates. We examined the arginine symmetric dimethylation of RORα using an anti-symmetric dimethyl-arginine (SDMA) antibody in immunoprecipitated RORα-Flag protein by using anti-Flag M2 beads. Co-transfection with PRMT5 dramatically enhanced the arginine symmetric dimethylation of the RORα (Figure 2a). We also performed PRMT5 protein pull-down using the SDMA antibody and further confirmed that the RORα was pulled down only in the PRMT5-overexpressing cells (Figure 2b).

As a typical nuclear receptor, the RORα domain structure consists of three major domains: an N-terminal highly conserved DNA-binding domain (DBD), a ligand-binding domain (LBD), and a hinge domain spacing the DBD and LBD domains of the RORα1 isoform. To further understand which domain of RORα is symmetrically dimethylated by PRMT5, we made different types of RORα protein clones. RORα1 (R1) and RORα4 (R4) are different isoforms of RORα, differing in terms of their N-terminal region. We found that PRMT5 enhanced both RORα1 and RORα4 arginine symmetric dimethylation, and it mainly symmetrically dimethylated the DBD domain of RORα but not the LBD domain (Figure 2c). 

Arginine methylation is a ubiquitous post-translational modification. We hypothesized that PRMT5 regulates RORα protein stability by arginine symmetric dimethylation modification. We performed a time-course experiment to determine RORα protein stability in control, PRMT5-overexpressed, and PRMT5-silenced cells with a cycloheximide treatment. The results showed that overexpression of PRMT5 significantly enhanced RORα protein stability, while silencing PRMT5 significantly accelerated RORα protein degradation (Figure 2d,e).

### 3.3. Knockdown of PRMT5 Decreases RORα Expression and Enhances Epithelial–Mesenchymal Transition (EMT) 

To assess the functional significance of decreased PRMT5 expression in the EMT process, we used two shRNA lentiviruses to silence PRMT5 expression (Figure 3a,b). The RORα protein level was decreased in the PRMT5-silenced cells (Figure 3b). RORα suppresses EMT by directly regulating the transcription of the key EMT inducer Snail [10]. We found increased *SNAI* mRNA levels in the PRMT5-silenced cells (Figure 3c). We also examined the EMT phenotypes in those cells. In the 2D culture, the PRMT5-silenced MCF10A cells exhibited an enhanced EMT phenotype (Figure 3d). In the in vitro 3D Matrigel culture system, the PRMT5-silenced MCF10A cells displayed an aggressive phenotype characterized by invasive branching, contracting with the organized sphere structures formed by the vector-control-infected MCF10A cells (Figure 3d). Additionally, the expression of the epithelial cell marker E-cadherin was diminished in the PRMT5-silenced MCF10A cells (Figure 3e). In contrast, the expression of the mesenchymal cell marker Snail was upregulated, while Vimentin tended to increase upon PRMT5 knockdown (Figure 3e). These data demonstrate that reduced PRMT5 expression downregulates RORα expression accompanied by increased Snail expression and enhanced EMT phenotypes in mammary epithelial cells.

### 3.4. Knockdown of PRMT5 Enhances Cell Invasion and Migration 

EMT is associated with the induction and maintenance of cell stemness properties. We examined cell stemness properties by introducing mammosphere formation in a non-attached cell culture plate with a stem cell medium. The mammosphere formation was dramatically increased in the PRMT5-silenced MCF10A cells (Figure 4a,b). The role of EMT in advancing caner is widely recognized, as it enhances cell invasion and migration abilities and consequently promotes cancer progression and metastasis. Consistently, we found that cell invasion was increased in the PRMT5-silenced MCF10A cells (Figure 4c,d). We also tracked single-cell migration using a live cell/incubator imaging system. Silencing of PRMT5 expression in the MCF10A cells significantly enhanced their cell migration ability (Figure 4e). These results suggest that reduced PRMT5 expression promotes cell stemness properties and enhances cell invasion and migration abilities in mammary epithelial cells. 

### 3.5. Overexpression of PRMT5 Elevates RORα Expression and Suppresses Cell Growth and Invasion 

To determine whether exogenous PRMT5 expression can elevate the RORα protein level and suppress malignant phenotypes in mammary epithelial cells, we introduced exogenous PRMT5 into MCF10A cells (Figure 5a,b). The Western blot analysis result showed that the RORα protein level was increased in the PRMT5-overexpressed cells (Figure 5b). In the 3D Matrigel culture system, the PRMT5-overexpressed MCF10A cells formed smaller spheres compared with the control group cells (Figure 5c,d). Consistently, we found that cell invasion was significantly decreased in the PRMT5-overexpressed MCF10A cells (Figure 5e,f). These results suggest that PRMT5 overexpression elevates RORα protein levels and suppresses the 3D growth and cell invasion ability of mammary epithelial cells.

## 4. Discussion

RORα has been reported as a potential tumor suppressor, and its expression is downregulated in almost all solid tumors. RORα expression is regulated at multiple levels. At the genomic level, the RORα gene is located at the middle of common fragile sites (15q22.2), large genomic regions that are highly unstable and prone to breakage and rearrangement, especially in cancer cells [28]. At the transcriptional level, a circular RNA, circGSK3B, is reported to regulate the binding of EZH2 and H3K27me3 to the RORα promoter [29]; androgen and its receptor, androgen receptor, are involved in the suppression of RORα expression, while estrogen and estrogen receptor are involved in the upregulation of RORα expression [30]. RORα protein is also regulated by various post-translational modifications, such as phosphorylation by protein kinase C [31], SUMOylation by SUMO-1 and SUMO-2 [32], or mono-methylation by EZH2 at lysine 38 [33]. RORα methylated by EZH2 can be recruited and ubiquitinated by DCAF1, which results in the proteasomal degradation of RORα [33]. In our study, the results showed that PRMT5 expression silenced by two different shPRMT5 lentiviruses can downregulate RORα expression, while overexpressed PRMT5 can upregulate RORα protein levels in mammary epithelial cells. Our findings in this study unveil a new post-translational modification for RORα, i.e., symmetric dimethylation of arginine mediated by PRMT5, which can enhance RORα protein stability by increasing the half-life of RORα. 

PRMT5, which belongs to the methyltransferase family, is a type II methyltransferasec that catalyzes the symmetric dimethylation of the amino acid arginine in target proteins [18,19,20]. The symmetric dimethylation of arginine by PRMT5 is also an important post-translational modification for the regulation of protein stability/half-life. For example, PRMT5 regulated HIF1α protein expression and stability during hypoxia [21]. MYC protein is reported to be symmetrically dimethylated by PRMT5 and asymmetrically dimethylated by PRMT1, which both regulate MYC stability and activity [22]. SREBP1, a crucial regulator of lipid accumulation and desaturation, undergoes symmetric dimethylation by PRMT5, thereby stabilizing it in clear-cell renal cell carcinoma [23]. PRMT5 methylates KLF4 and inhibits its ubiquitylation and consequently enhances KLF4’s half-life, which results in elevated KLF4 protein expression levels [25]. Furthermore, in breast cancer cells, PRMT5 methylates KLF5 and inhibits its phosphorylation, ubiquitylation, and degradation [34]. In our study, we demonstrated that PRMT5 stabilizes the RORα protein mainly through the symmetric dimethylation of its highly conserved DBD domain. Therefore, one of our future investigation priorities is to determine the specific arginine sites of the DBD domain, which is symmetrically dimethylated by PRMT5. This will involve a combination of bioinformatics analysis and experimental validation to examine the symmetric dimethylation sites by using site-mutated or truncated DBD-expressing vectors.

PRMT5, identified as an oncogene, demonstrates elevated levels across various cancer types, including lung cancer, melanoma, glioma, prostate cancer, and lymphoma. Elevated PRMT5 expression is associated with poor clinical outcomes in cancer patients [35,36,37,38,39,40]. Several PRMT5 inhibitors are currently under clinical trial studies for cancer therapy [41,42,43]. In breast cancer, several investigations have shown that PRMT5 is elevated in its expression level in breast cancer cells and is correlated with a poor survival ratio in breast cancer patients, and it promotes cell invasion and proliferation and inhibits cell apoptosis in breast cancer cells [26,44,45,46]. However, it is reported that PRMT5 protein expression remains comparable among normal mammary epithelial cells, luminal breast cancer cells, and aggressive triple-negative breast cancer cells (TNBCs) [26]. Moreover, studies have indicated that tamoxifen induces PRMT5 translocation to the nucleus in tamoxifen-sensitive cells, with higher nuclear PRMT5 levels associated with improved survival in luminal breast cancer cells [27,47]. PRMT5 predominantly localizes in the cytoplasm in aggressive TNBCs, while it tends to localize in the nucleus of certain luminal breast cancer cells and normal mammary epithelial cells [26,27]. The subcellular localization of PRMT5 significantly affects its function in cancer cells. For instance, PRMT5 exhibits divergent roles in the cytoplasm and nucleus of prostate cancer cells. While nuclear PRMT5 inhibits cell growth in the benign prostate epithelium, cytoplasmic PRMT5 is essential for cancer cell growth in premalignant and prostate cancer tissues [38]. In lung cancer, high levels of cytoplasmic PRMT5 are correlated with a higher grade and poorer prognosis in aggressive non-small-cell lung carcinomas, whereas nuclear PRMT5 is more frequently detected in non-aggressive carcinoid tumors [48]. Similarly, metastatic melanomas exhibit reduced nuclear PRMT5 expression compared to primary cutaneous melanomas, with higher cytoplasmic PRMT5 expression observed in melanoma tissue relative to normal skin epidermal cells [36]. In the future, it will be important to examine the correlation between nuclear PRMT5 and RORα in human breast cancer tissue.

## 5. Conclusions

Here, we found that in mammary epithelial cells, PRMT5 is reported to be mostly localized in the nucleus [26,27], where PRMT5 symmetrically dimethylates and stabilizes the transcriptional factor RORα protein level. A high expression of RORα helps to maintain the differentiation of mammary epithelial cells. When PRMT5 is silenced in mammary epithelial cells, RORα protein is degraded and downregulated, and it consequently promotes mammary epithelial cells into transforming and initiating breast cancer. In conclusion, we identified a novel mechanism for RORα regulation through PRMT5-dependent symmetric dimethylation. It is important to determine whether this pathway contributes to the downregulation of RORα during breast cancer development and progression in the future.

## Figures and Tables

**Figure 1 cancers-16-01914-f001:**
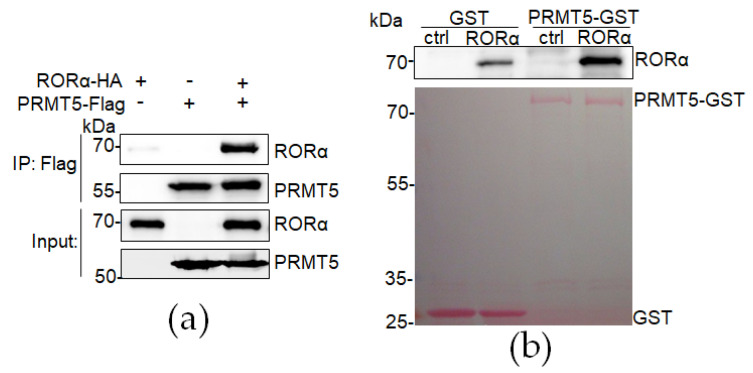
RORα directly interacts with PRMT5. (**a**) RORα protein was immunoprecipitated by PRMT5-Flag protein using anti-Flag antibody in Co-IP experiment; (**b**) binding of RORα protein with PRMT5-GST protein detected by GST pull-down experiment.

**Figure 2 cancers-16-01914-f002:**
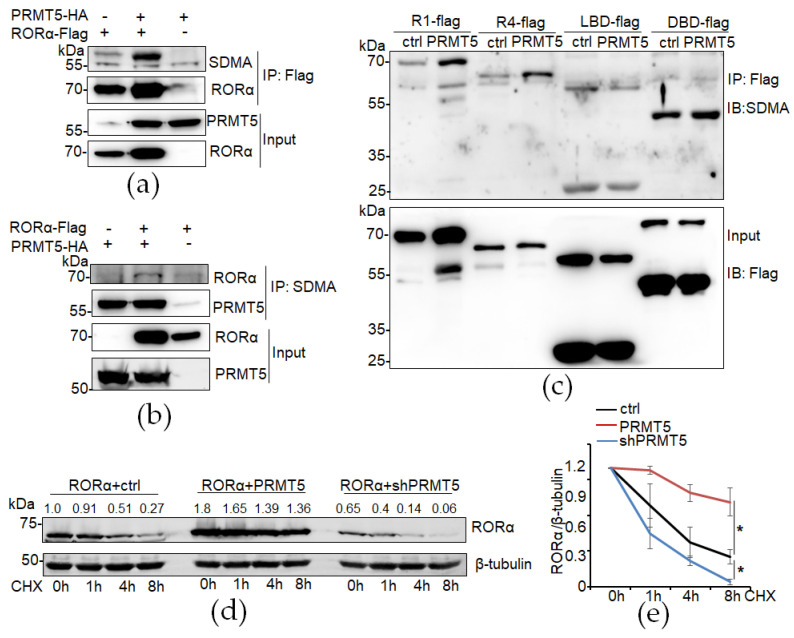
RORα is symmetrically dimethylated and stabilized by PRMT5. (**a**,**b**) Increased arginine symmetric dimethylation of RORα in PRMT5-expressing cells immunoprecipitated using anti-Flag antibody; immunoprecipitated anti-symmetric dimethyl-arginine (SDMA) antibody. (**c**) DBD domain of RORα was symmetrically dimethylated by PRMT5. (**d**,**e**) Analysis of RORα protein degradation in control, PRMT5, and shPRMT5-transfected HEK293FT cells with cycloheximide (CHX, 100 μM) treatment. Results are presented as mean ± SEM; *n* = 3. * *p* < 0.05, independent Student’s *t*-test.

**Figure 3 cancers-16-01914-f003:**
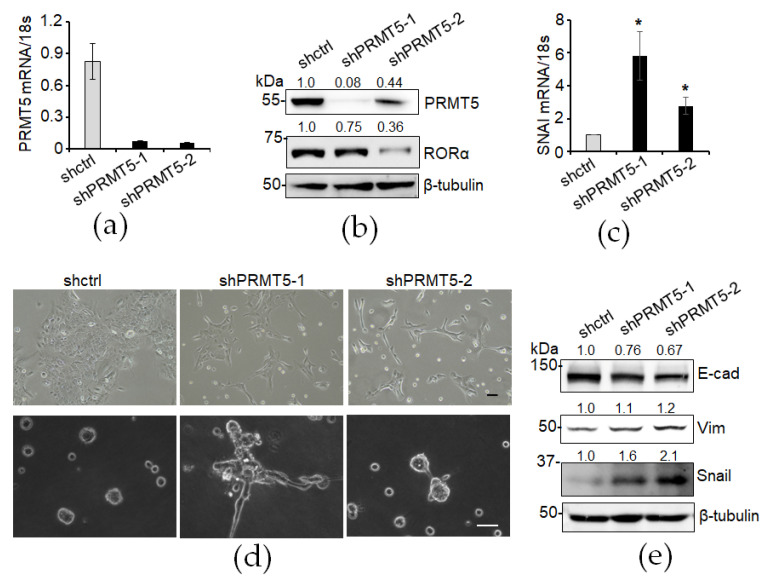
Silencing PRMT5 decreases RORα and promotes EMT. (**a**) PRMT5 knockdown was confirmed by RT-PCR; (**b**) PRMT5 knockdown was confirmed by Western blotting; (**c**) *SNAI* mRNA levels were increased in PRMT5-silenced MCF10A cells. Results are presented as mean ± SEM; *n* = 3. * *p* < 0.05, independent Student’s *t*-test. (**d**) Phase contrast images of control and two PRMT5-knockdown MCF10A cell lines (shPRMT5-1 and shPRMT5-2) in 2D and 3D cultures. Bar: 50 µm; (**e**) EMT marker proteins were examined in control and PRMT5-silenced MCF10A cells.

**Figure 4 cancers-16-01914-f004:**
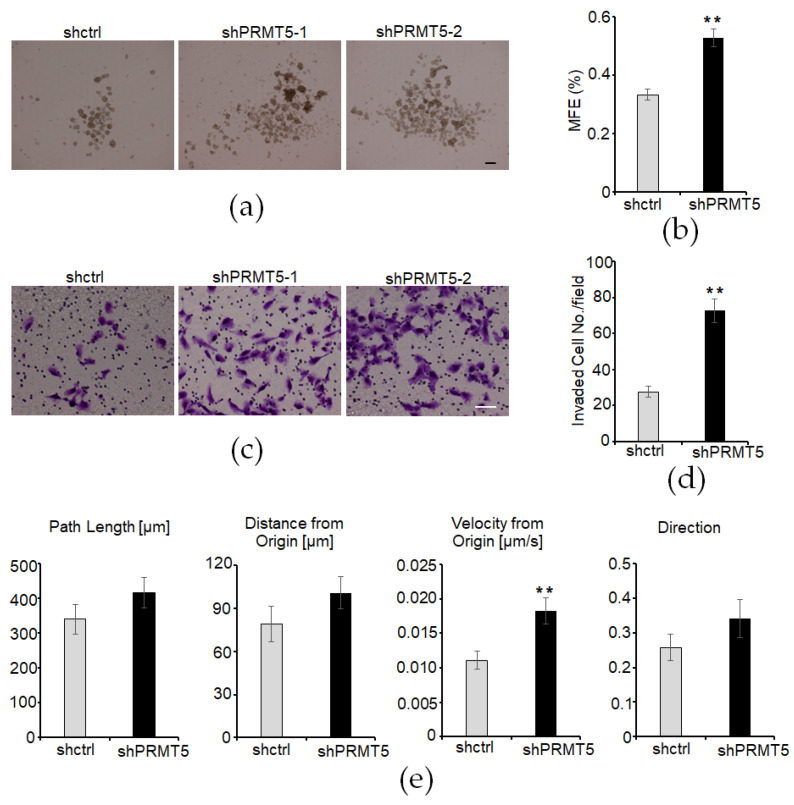
Silencing PRMT5 expression enhances cell invasion and migration. (**a**) Images of mammospheres in vector control or PRMT5-silenced MCF10A cells. Bar: 50 µm. (**b**) Mammosphere formation efficiency (MFE) was quantified in vector control or PRMT5-silenced MCF10A cells. Results are presented as mean ± SEM; *n* = 10; ** *p* < 0.01, determined by independent Student’s *t* test. (**c**) Transwell invasion analysis was conducted for both control and PRMT5-silenced MCF10A cells. Bar: 50 µm. (**d**) Invaded cells were quantified in vector control or PRMT5-silenced MCF10A cells. Results are presented as mean ± SEM; *n* = 3. ** *p* < 0.01, determined by independent Student’s *t* test. (**e**) Single-cell migration analysis was performed in control and PRMT5-silenced MCF10A cells. Results are presented as mean ± SEM; *n* = 100. ** *p* < 0.01, determined by independent Student’s *t* test.

**Figure 5 cancers-16-01914-f005:**
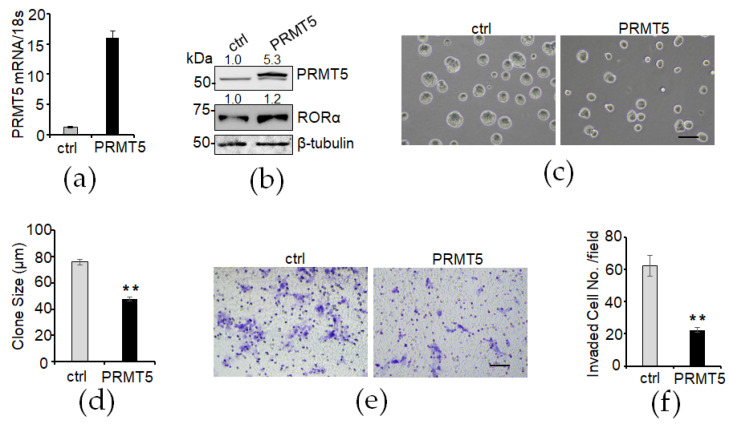
Elevated RORα protein levels in PRMT5-overexpressed cells suppresses clone growth and cell invasion. (**a**) PRMT5 overexpression was confirmed by RT-PCR; (**b**) PRMT5 overexpression was confirmed by Western blotting; (**c**,**d**) phase contrast images and quantification of control and PRMT5-overexpressed MCF10A cells in 3D culture. Bar: 100 µm. (**e**,**f**) Transwell invasion analysis of control and PRMT5-overexpressed MCF10A cells. Bar: 100 µm. Results are presented as mean ± SEM; *n* = 3. ** *p* < 0.01, independent Student’s *t* test.

## Data Availability

All data are contained within the article.

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
