# Peer review of "Regulation of RORα Stability through PRMT5-Dependent Symmetric Dimethylation"

_cancers, 2024, doi:10.3390/cancers16101914_

Round 1
Reviewer 1 Report
Comments and Suggestions for Authors
This manuscript reported a novel protein interaction between RORα and PRMT5 and RORα regulated by PRMT5 functional in cell invasion, stemness and EMT morphology including a change in Snail expression. However, most experiments were performed only in the MCF10A human breast epithelial cell line, which is the most commonly used normal breast cell model. Some experimental data should be validated in other breast cell lines and breast cancer cell lines. Many relevant published studies about PRMT5 in breast cancer cell lines and patients are not discussed or cited in the manuscript. For example, several breast cancer studies are analyzed in this literature “PRMT5 was pan-cancerous as a prognostic biomarker, and high level of PRMT5 was associated with poor prognosis for certain cancers. Clinicopathological and Prognostic Significance of PRMT5 in Cancers: A System Review and Meta-Analysis. Liang Z, Liu L, Wen C, Jiang H, Ye T, Ma S, Liu X.Cancer Control. 2021 Jan-Dec;28:10732748211050583. doi: 10.1177/10732748211050583.PMID: 34758643 “.
Author Response
Dear Reviewer,
We greatly appreciate your thoughtful comment regarding most of experiments performed only in MCF10A cells, and your suggestion for inclusion of validated experimental data in other breast cell lines and breast cancer cell lines. In our study, we have investigated the interaction between PRMT5 and RORα, and examined PRMT5 stabilizes RORα protein by dimethylation in HEK293FT cells, and further validated PRMT5 regulates both the protein level and function of RORα in MCF10A cells. Our study focuses on elucidating a new post-translational modification for the tumor suppressor RORα. We aim to understand the mechanism underlying the downregulation of RORα in breast cancer cells as described in our prior publications. We guess that translocation of PRMT5 from nuclear in normal mammary epithelial cells to the cytoplasmic in breast cancer cells, leads to decreased demethylation of RORα, a nuclear receptor, consequently resulting in the downregulation of RORα protein levels. We will introduce more cell lines to validate our hypothesis and examine the correlation between nuclear PRMT5 and RORα in human breast cancer tissue in the future.
Thanks for your reminder to discuss several relevant published studies about PRMT5 in breast cancer cell lines and patients. In the revised manuscript, we added the citation of the literature discussed PRMT5 as prognostic biomarker for pan cancers and associated with poor prognosis for certain cancers (See line 368, refer 40 in revised manuscript) and discussed the relevant published studies analyzed within this literature (See line 369-372 in revised manuscript).
Reviewer 2 Report
Comments and Suggestions for Authors
The manuscript unveils a novel mechanism by which PRMT5 regulates RORα protein stability, shedding light on the intricate molecular pathways involved in breast cancer development and progression. The study suggests that targeting PRMT5-mediated regulation of RORα could have therapeutic implications for breast cancer treatment. This is an interesting manuscript. But I have several following concerns:
1. This article has a 42% duplication rate with the reported literature, and even a single article has more than 8%, which is unacceptable.
2. Abbreviations should be defined when they first appear in the text. Such as "cDNA", "LBD", "DBD" in Line 101,"EGF" and "FBS" in Line 176, ...
3. In Line 169, the "5" of "1 × 105" should be superscript.
4. Please mark the size of each protein in each picture and quantify it in gray scale, so that readers can understand the size and change of protein more intuitively.
5. Please add a scale bar for the Figures in the text.
6. The authors considered that PRMT5 regulates the stability of RORα by symmetric dimethylation, and it is best to determine the site of methylation modification by bioinformatics experiments and site-mutation related assays.
7. The nucleic acid sequences (including gene names, regulatory sequences, and primer names) should be in italics.
8. Please unify the format of references in the article, including the author's name, the case of words in the title of the article, the writing of the name of the journal, and the page number.
Comments on the Quality of English LanguageMinor editing of English language required
Author Response
Dear Reviewer:
We thank you considering our study as shedding light on the intricate molecular pathways involved in breast cancer development and progression. We are encouraged by you recognition of our research as interesting. We also appreciated your very thoughtful and constructive comments, which have been most helpful in refining and strengthening this study.
Here is our reply to your comments:
- This article has a 42% duplication rate with the reported literature, and even a single article has more than 8%, which is unacceptable.
Response: Thank you for your reminder. We have revised the some of the introduction and discussion parts in our revised manuscript.
- Abbreviations should be defined when they first appear in the text. Such as "cDNA", "LBD", "DBD" in Line 101,"EGF" and "FBS" in Line 176, ...
Response: We have added full names for the abbreviations in the revised manuscript (See line 83, 87, 102, 103, 121, 122 in revised manuscript).
- In Line 169, the "5" of "1 × 105" should be superscript.
Response: Thank you for your good catch. We have corrected this in the revised manuscript (See line 174 in revised manuscript).
- Please mark the size of each protein in each picture and quantify it in gray scale, so that readers can understand the size and change of protein more intuitively.
Response: Thank you for your advice. We have marked the size of each protein in each picture (See Figure 1, 2, 3, and 5) and added the quantifications in most of the WB results (See Figure 2, 3, and 5).
- Please add a scale bar for the Figures in the text.
Response: Thanks for your reminder, we have added scale bar in the revised Figure 3, 4, and 5 and figure legends (See line 286, 304, 306, 325, and 326).
- The authors considered that PRMT5 regulates the stability of RORα by symmetric dimethylation, and it is best to determine the site of methylation modification by bioinformatics experiments and site-mutation related assays.
Response: Thank you very much for the constructive suggestion. Identify the specific sites of RORα for PRMT5 dimethylation is crucial for our study and this is what we plan to do in the future. We have discussed this in revised manuscript (See line 359-364 in revised manuscript).
- The nucleic acid sequences (including gene names, regulatory sequences, and primer names) should be in italics.
Response: We have corrected the nucleic acid sequences and primer names in italics (See line 102, 103, 105, 106, 110-111, 168-170 in revised manuscript).
- Please unify the format of references in the article, including the author's name, the case of words in the title of the article, the writing of the name of the journal, and the page number.
Response: Thank you for reminder, we have downloaded the MDPI reference style and corrected the format of references.
Reviewer 3 Report
Comments and Suggestions for Authors
Authors demonstrated the role of PRMT5-RORα axis in a breast cancer cell line. Several in vitro experiments were performed to support the relevance of PRMT5-RORα interplay in breast cancer cells. The following points have to be addressed before publication:
- In legend of Figure 2 specify the meaning of SDMR.
- I understand that you performed two shRNA lentivirus transduction. Which are the differences between both transductions? Are they replicates? Figure 3b-d shows differences between shPRMT5-1 and shPRMT5-2. Is it a transduction efficiency issue? Please, specify it and discuss it in the Discussion section.
- In Figure 3E Vimentin expression is almost similar among conditions. In line 271 you say that Vimentin is upregulated. Is this increased expression significant? If it is significant please specify it, otherwise I suggest changing the phrase as follows: Vimentin expression tends to increase.
- Is the Figure reference in line 312 correct? Invasion assays are letter e and f.
- The first paragraphs of the discussion are a description of literature research without any comparison and real discussion of the manuscript results. In the last paragraph you only mention the obtained results. Please, rewrite the discussion.
- The statement of lines 50-52 is not clear. A verb or punctuation is missing. Please, rewrite it.
- In legend of Figure 1 letter a is missing.
Comments on the Quality of English LanguageMinor editing of English is required as suggested in the comments.
Author Response
Dear Reviewer:
We thank you considering our data supportive of the relevance of PRMT5-RORα interplay in breast cancer cells. Additionally, we greatly appreciated your very thoughtful and constructive comments, which have been instrumental in refining and strengthening this study.
Here is our reply to your comments:
- In legend of Figure 2 specify the meaning of SDMR.
Response: Thank you for reminder, we have corrected the “SDMR” to “SDMA” in the revised manuscript (See line 261).
- I understand that you performed two shRNA lentivirus transduction. Which are the differences between both transductions? Are they replicates? Figure 3b-d shows differences between shPRMT5-1 and shPRMT5-2. Is it a transduction efficiency issue? Please, specify it and discuss it in the Discussion section.
Response: These two shRNAs are two different clones targeting different sequences of PRMT5. We have added detail description in “Materials and Methods” (See line 108-111) and “Discussion” (See line 341-343).
- In Figure 3E Vimentin expression is almost similar among conditions. In line 271 you say that Vimentin is upregulated. Is this increased expression significant? If it is significant please specify it, otherwise I suggest changing the phrase as follows: Vimentin expression tends to increase.
Response: Thank you for your advice. We have corrected this description in revised manuscript (See line 277-278).
- Is the Figure reference in line 312 correct? Invasion assays are letter e and f.
Response: Thank you for your nice catch. We have corrected this in revised manuscript (See line 318).
- The first paragraphs of the discussion are a description of literature research without any comparison and real discussion of the manuscript results. In the last paragraph you only mention the obtained results. Please, rewrite the discussion.
Response: Thank you for your comment. We have rewritten the first paragraph (See line 341-346) in revised manuscript. As to the last paragraph of the discussion, we think it is better to be in “Conclusions” part (See line 392-397).
- The statement of lines 50-52 is not clear. A verb or punctuation is missing. Please, rewrite it.
Response: Thank you for your catch. We have rewritten this in revised manuscript (See line 50-53).
- In legend of Figure 1 letter a is missing.
Response: Thank you for your catch. We have added letter “a” this in revised manuscript (See line 232).
Round 2
Reviewer 1 Report
Comments and Suggestions for Authors
It is acceptable now.
Author Response
We would like to expression our gratitude to the reviewer for considering our manuscript acceptable for publication.
Reviewer 2 Report
Comments and Suggestions for Authors
The authors have addressed all my comments, I recommend accepting it in current form.
Author Response
We would like to expression our gratitude to the reviewer for satisfying with our revised manuscript and considering it acceptable for publication.